**Data Availability Statement:** The public data sharing is under the e-clearance procedure. Data will be available through the Alliance for Public Health repository (URL: http://aph.org.ua/en/resources/publications/). The dataset and

# HIV treatment cascade among people who inject drugs in Ukraine

Yana Sazonova[1]☉*, Roksolana Kulchynska[2]☉, Yuliia Sereda[3]‡, Marianna Azarskova[2]‡, Yulia Novak[1]‡, Tetiana Saliuk[1]‡, Marina Kornilova[1]‡, Mariia Liulchuk[4]‡, Charles Vitek[5]‡, Kostyantyn Dumchev[6]☉

1 Monitoring and Evaluation Unit, ICF "Alliance for Public Health", Kyiv, Ukraine, 2 Division of Global HIV and TB, U.S. Centers for Disease Control and Prevention, Kyiv, Ukraine, 3 Independent Researcher, Kyiv, Ukraine, 4 State Institution "The L.V. Gromashevskij Institute of Epidemiology and Infectious Diseases of NAMS of Ukraine", Kiev, Ukraine, 5 Division of Global HIV and TB, U.S. Centers for Disease Control and Prevention, Atlanta, GA, United States of America, 6 Ukrainian Institute on Public Health Policy, Kyiv, Ukraine

☉ These authors contributed equally to this work.
‡ These authors also contributed equally to this work.
* yana.sazonova2@gmail.com

## Abstract

The HIV treatment cascade is an effective tool to track progress and gaps in the HIV response among key populations. People who inject drugs (PWID) remain the most affected key population in Ukraine with HIV prevalence of 22% in 2015. We performed secondary analysis of the 2017 Integrated Bio-Behavioral Surveillance (IBBS) survey data to construct the HIV treatment cascade for PWID and identify correlates of each indicator achievement. The biggest gap in the cascade was found in the first "90", HIV status awareness: only 58% [95% CI: 56%-61%] of HIV-positive PWID reported being aware of their HIV-positive status. Almost 70% [67%-72%] of all HIV-infected PWID who were aware of their status reported that they currently received antiretroviral therapy (ART). Almost three quarters (74% [71%-77%]) of all HIV-infected PWID on ART were virally suppressed. Access to harm reduction services in the past 12 months and lifetime receipt of opioid agonist treatment (OAT) had the strongest association with HIV status awareness. Additionally, OAT patients who were aware of HIV-positive status had 1.7 [1.2–2.3] times the odds of receiving ART. Being on ART for the last 6 months or longer increased odds to be virally suppressed; in contrast, missed recent doses of ART significantly decreased the odds of suppression. The HIV treatment cascade analysis for PWID in Ukraine revealed substantial gaps at each step and identified factors contributing to achievement of the outcomes. More intensive harm reduction outreach along with targeted case finding could help to fill the HIV awareness gap among PWID in Ukraine. Scale up of OAT and community-level linkage to care and ART adherence interventions are viable strategies to improve ART coverage and viral suppression among PWID.

supporting study information are currently available with the request to the Alliance for Public Health (salyuk@aph.org.ua, Principal Investigator) or Public Health Center of Ministry of Health of Ukraine (info@phc.org.ua).

**Funding:** IBBS study was supported by the U.S. President's Emergency Plan for AIDS Relief (PEPFAR) through the U.S. Centers for Disease Control and Prevention under the terms of grant NU2GGH000840 (Engaging Local Indigenous Organizations in Developing HIV/AIDS Monitoring and Evaluation Capacity in Ukraine). The findings and conclusions in this paper are those of the authors and do not necessarily represent the official position of the funding agencies. The authors received no specific funding for this publication.

**Competing interests:** The authors have declared that no competing interests exist

# Introduction

Due to its dual impact on mortality and transmission, antiretroviral therapy (ART) for HIV has become a top priority in the global HIV response. UNAIDS set ambitious 90-90-90 HIV treatment targets to facilitate HIV case finding, scale-up treatment, and to improve treatment outcomes by 2020 [1].

The HIV treatment cascade is an increasingly popular tool to identify and visualize achievements and gaps in progress towards the 90-90-90 targets. UNAIDS estimates that globally in 2017 about 70% (CI: 51–84%) of people living with HIV (PLWH) knew about their HIV-positive status, of them 77% (CI: 57–89%) received ART treatment, and 82% (CI: 60–89%) of those on treatment were virally suppressed [2]. In Ukraine, the gaps in the national cascade are more substantial: an estimated 58% of PLWH are aware of their HIV status, 63% receive ART and 69% have undetectable viral load [3].

People who inject drugs (PWID) in Eastern Europe have high HIV prevalence and continue to play a key role in HIV epidemics [4–7]. Data on PWID cascade is scarce [8]. In countries where data on PWID are available, disproportionately poorer outcomes are reported for this key population at each step of the continuum, presenting distinct challenges in achieving the 90-90-90 targets [9, 10].

Multiple studies have investigated the mechanisms and factors that contribute to the obstacles in accessing HIV testing and treatment services. Awareness of an HIV-positive status is lower among PWID of younger age, male sex, lower education, absence of a regular sexual partner, unemployment, high frequency of injection drug use (IDU), injection of amphetamines, more frequent alcohol consumption, and selling sex [11–13]. PWID with high frequency of drug and alcohol use, those having commercial sexual partners, incarceration experience or substantial housing instability were less likely to receive ART and achieve viral suppression [14–18].

On the contrary, access to HIV prevention services or receiving treatment services related to sexually transmitted infections or drug addiction were found to be positively associated with HIV status awareness [11, 13]. Tuberculosis screening and treatment services were found to be factors that increase ART intake, and opioid agonist treatment (OAT) has increased odds for both ART intake and viral suppression [14, 19–21].

The routine HIV surveillance systems rarely capture socio-demographic and behavioral risk factors that may predict outcomes along the HIV treatment continuum. Such data can be collected using population-based surveys, especially among key populations, such as integrated bio-behavioral surveys (IBBS), conducted in many countries around the globe [22, 23]. PWID in Ukraine remain the most affected key population with an HIV prevalence of 22% in the 2015 IBBS [24]. IBBS among PWID have been conducted biennially since 2007 [25]. The 2017 IBBS round included testing for HIV viral load, for the first time providing sufficient data to assess all of the indicators on the HIV treatment continuum. The aim of this analysis was to estimate HIV treatment cascade indicators among PWID in Ukraine and to identify correlates of key outcomes based on the 2017 IBBS data.

# Materials and methods

## Study setting and design

We used data from the 2017 national IBBS among PWID, implemented by the Alliance for Public Health in September-December 2017. Details on methodology and results of previous IBBS rounds in Ukraine are available elsewhere [24–30].

Survey sites were established in 30 cities of Ukraine, including all capital cities and six smaller cities of 23 states; and one city in the Crimea Autonomous Republic. Each city had one study site.

Study sites were selected based on prior formative assessments to ensure comfort and safety for participants. Selection excluded locations in close proximity to HIV service providers (non-government organizations (NGO) working in HIV/AIDS field, HIV clinics or ART sites) to minimize the overrepresentation of harm reduction clients or ART patients in the sample.

## Sample size and recruitment

Sample size was calculated for each study site based on the PWID population size, the city-level HIV prevalence measured in the previous 2015 IBBS [24], and the desired precision level (0.05), taking into account the design effect. City samples ranged from 200 to 550 PWID. In total, 10,076 PWID were recruited.

Participants were recruited using Respondent-Driven Sampling (RDS) approach [31–35]. In each city, the survey started with two to four initial participants (seeds) depending on the planned sample size. Seeds were purposefully selected according to demographic and behavioral characteristics reflecting diversity in PWID networks. Once identified, the seeds each recruited three PWID from their social network. If a seed failed to recruit study participants, they were substituted with another one with similar characteristics.

Each enrolled participant was offered three coupons to recruit PWID from their social network. Inclusion criteria for subsequent participants were age of 14 years or older (self-reported), injecting drugs in the last 30 days (verified by presence of visible injection marks), residence in the city of recruitment, and not participating in any other surveys during the last six months (self-reported). Each participant was asked to give informed consent prior to participation in the survey. Study participants received the equivalent of 6 USD as compensation for their time and travel. In addition, recruiters received the equivalent of 1.5 USD for each referred peer who qualified for the study. The participants' compensation was determined during the formative assessment and was approved by an Institutional Review Board. The amount was deemed adequate for the required scope of assessment, and neither coercive nor demotivating for the majority of the potential study participants.

## Data collection

Participants IDs were generated with unique combinations of letters and numbers that were used for QR codes printed on RDS recruitment coupons. Personally identifiable information was not used to generate IDs.

Information on social and demographic characteristics, drug use and risky injection practices, sexual behaviors, use of HIV prevention services, experience of HIV testing and treatment, treatment of drug abuse were collected using face-to-face interviews. All data were recorded using the IBBS module of the SyrexCloud mobile application that was specifically developed for this survey [36].

After the interview, trained healthcare workers performed testing for HIV and antibody to hepatitis C virus (anti-HCV) using rapid tests. HIV status was determined using the three rapid test algorithm for populations with HIV prevalence more than 5% according to the World Health Organization (WHO) guidelines [37]. Profitest test HIV 1/2 was the first test, SD Bioline HIV 1/2 3.0 was the second test and Alere Determine™ HIV-1/2 was the third.

Testing for anti-HCV was done using HCV Profitest rapid test.

Dry blood spots specimens for viral load determination were collected from all participants who tested positive for HIV. Viral load testing was performed using Abbott m2000rt Real Time HIV test in the national virology reference laboratory [38].

Pre- and post-test counseling was provided to all participants. During the post-test counselling, medical workers asked the participants about previous knowledge of their HIV-status

and ART intake. These data were added to the main dataset to cross-check with the interview results. HIV-positive participants were referred to case-management services to expedite enrollment to HIV care.

The procedures of data quality assurance were implemented throughout the data collection process. To ensure the protocol and SOPs compliance, site monitoring visits were conducted and intense supervision was maintained. To exclude overrepresentation or underrepresentation of some groups the RDS recruitment process was checked weekly for recruitment and population homophily for the main socio-demographic characteristics, harm reduction and HIV testing services utilization, HIV-prevalence, and HIV treatment cascade indicators.

## Cascade outcomes and correlates

Our choice and definition of indicators was determined by WHO practice guidelines [39], Ukrainian healthcare system, and the PWID-specific context.

Only participants who tested positive by the HIV testing algorithm were included in the cascade analysis. Then we developed three outcome variables: awareness of HIV status, ART uptake, and viral suppression.

We defined the awareness of HIV positive status based on self-reported knowledge of a previous HIV positive test result, regardless of when and where it was performed. If participants did not report their positive status during the interview, but later reported it to the medical worker during post-test counselling, they were also considered to be aware about HIV-positive status. Those participants who refused to report status and did not report it in post-test counseling were coded as missing values and excluded from the analysis. We performed sensitivity analysis to investigate whether the participants who refused to report their status differed in their characteristics and whether their exclusion could affect our results.

The next outcome (ART uptake) was assessed by self-report among those HIV-positive participants who were aware about HIV-positive status. It was determined by one question *"Are you currently on antiretroviral therapy?"*.

Amongst the participants who answered positively to the ART uptake question, viral suppression outcome was defined as a viral load of <1000 HIV RNA copies per mL.

To construct an HIV treatment cascade, we used a sequential approach to present achievement of 90-90-90 targets. For each cascade bar, we present a proportion based on the preceding indicator in the cascade (as implied in the 90-90-90 targets).

The following variables were examined as potential correlates of selected outcomes.

**Socio-demographic variables.** We included age, sex, education, marital status, employment, living conditions during the last three months (unstable or homeless living conditions defined as living at the street, abandoned apartments, railway stations or frequent moving from one place to another compared to other), monthly income (below or above the standard minimum wage in Ukraine in 2017), duration of injection drug use in years, recent (during last year) and lifetime incarceration history.

**Behavioral variables.** Risky injection behavior was defined as having at least one of the following practices in the last 30 days: receptive needle/syringe sharing, cooking or distribution of drugs with common instruments, or purchasing drugs in the prefilled syringe. Each reported type of injected drug that have been used in the last 30 days was grouped into three categories: category of exclusive opioid injecting drug use (e.g. heroin, opium, desomorphine, home-made opioids, illegal methadone, or buprenorphine), category of exclusive stimulant injecting drug use (e.g. amphetamines, methamphetamines, cocaine, salt), and category of mixed injecting drug use. Injection frequency was defined as the number of days of drug

injecting in the past 30 days. Risky sexual behavior included having any commercial sex (selling or buying) in the last 90 days.

**Health seeking variables.** We also defined several variables describing utilization of health services: accessing harm reduction programs (at least once in the past 12 months), receiving any non-HIV-specific medical services (e.g. primary care or specialized care for non-HIV related conditions in the past 12 months), tuberculosis treatment in the past 12 months, receiving OAT (at the time of the survey or ever in lifetime).

The result of anti-HCV testing in the survey was included into the model as a separate factor, as well as self-reported information about lifetime syphilis diagnosis.

Additionally, we included two variables describing the time since ART prescription, and current ART non-adherence (defined as missing ART doses for the past two days) as potential predictors of viral suppression.

**Contextual variables.** To account for city-level factors in the multilevel models, we selected variables that could contribute to the HIV cascade outcomes. These included the estimated number of PWID living with HIV, and the number of HIV rapid tests performed by harm reduction programs [40, 41]. Other contextual variables did not show significant association with outcomes and were therefore excluded from the final models, such as the number of harm reduction clients, ART coverage in the city, number VL tests, average CD4 count among PWID at the time of enrollment into HIV care, average first VL test result among PWID [42].

## RDS diagnostics

RDS assumptions were tested for the main outcomes (HIV status awareness, self-reported ART uptake and VL suppression) in each city using convergence and bottleneck plots, and recruitment and population homophily in RDS-Analyst software [43, 44]. Convergence estimates, including HIV status awareness, self-reported ART uptake and VL suppression, stabilized after the half of observations in the most of the study cities. Bottleneck plots have shown that HIV status awareness, self-reported ART uptake and VL suppression estimates were similar products in most of the cities among the recruitment chains. Recruitment homophily was acceptable in all study cities: recruitment homophily of HIV status awareness ranged from 0.98 (Odesa city) to 1.32 (Chernihiv city); recruitment homophily of self-reported ART uptake ranged from 0.98 (Odesa city) to 1.29 (Chernihiv city), recruitment homophily of VL suppression ranged from 0.98 (Odesa city) to 1.29 (Chernihiv city) (S1 Table).

## Statistical analysis

Each city dataset was weighted using imputed visibility procedure calculated in RDS-Analyst version 0.72 [43]. All results were adjusted using RDS-Analyst weights coefficients that consider the RDS design and population size estimation for each city. Then, we used aggregation function to merge all RDS-Analyst weights coefficients. The pooled dataset for 30 cities was analyzed in R to produce aggregate cascade estimates (version 3.5.1) [45].

First, we conducted bivariate analysis to assess each outcome variable of the cascade in disaggregation by socio-demographic, behavioural and health-seeking variables. We compared categorical variables using chi-square test. T-test was used to assess significance of relationship of outcomes and continuous variables.

To identify correlates for each cascade outcomes, accounting for nested structure of the data (participants within cities), and multilevel models were used.

Links between selected covariates and outcomes were measured with a generalized linear mixed model (GLMM) by maximum likelihood, logit link function. First, null models (no predictors) with random intercepts were fitted and Variance Partition Coefficients (VPC) were

estimated to examine variance attributable to clustering within the city. Then, fixed effects for participant and city-level predictors as well as interactions between them were measured using backward elimination with Wald test [46]. Only significant (p<0.05) variables remained in multivariate multilevel analysis. All individual-level variables were examined for random effects using Likelihood Ratio Test (LRT). Unstructured covariance matrix was used for models with random predictors. The comparison models of goodness of fit included information criteria (Akaike Information Criterion (AIC), Bayesian information criterion (BIC)) and checking model assumptions. GLMM were estimated in R lmer4 package [47].

### Ethical approval

Prior to enrolment into the study, all participants were provided with comprehensive information about the study and signed a consent form. All study procedures were conducted according to the ethical standards of the institutional and national research committee and with the 1964 Helsinki declaration and its later amendments or comparable ethical standards.

The study was approved by Institutional Review Board (IRB) at the Ukrainian Institute on Public Health Policy (Kyiv, Ukraine). The study was also reviewed in accordance with the U.S. Centers for Disease Control and Prevention (CDC) human research protection procedures and determined to be research, but CDC investigators did not interact with human subjects or have access to identifiable data or specimens for research purposes.

## Results

### Cascade status

Of the total 10,076 participants were recruited in the IBBS, 2,261 tested HIV-positive, resulting in a 22.6% (95% CI: 21.8–23.4%) overall HIV prevalence. Socio-demographic and other characteristics of the participants by HIV status are presented in Table 1.

The HIV treatment cascade indicators are presented in Fig 1. Analysis of HIV status awareness was done in the sample of IBBS participants who were HIV-positive according to the testing algorithm and who agreed to report their HIV status during the interview or post-test counseling (n = 2,122). Of all those who tested positive, 6.1% (5.1–7.2%) refused to report previous knowledge about their HIV status and were excluded from the cascade analysis. Since ART initiation is only possible in those who are HIV-infected and are aware of status, analysis of ART uptake correlates was done in the respective subsample (n = 1,277); similarly, predictors of VL suppression were analyzed in the subsample of participants who reported receiving ART (n = 895), excluding the small number of participants (n = 5) whose samples were rejected by the lab due to quality issues.

The biggest cascade gap was observed at the stage of HIV status awareness, with only 58% (95% CI: 56–60%) of HIV-infected PWID reporting having had a positive HIV test previously. 70% (95% CI: 67–72%) of the PWID who were aware about their HIV positive status reported receiving ART. 74% (95% CI: 71–77%) of PWID among ART patients had viral load less than 1000 cp/mL.

### Correlates of HIV treatment cascade outcomes

Table 2 presents the bivariate analysis of socio-demographic, behavioral and service uptake variables for each of the three HIV treatment cascade indicators. Analysis of socio-demographic variables showed that the probability of achieving each cascade outcome increased with age and duration of injection drug use for all three outcomes. Additionally, HIV status awareness was associated with being female and having a lower monthly income. ART uptake also was associated with lower monthly income. PWID with riskier injection behavior were

**Table 1. Distribution of socio-demographic and other characteristics in total sample and in disaggregation by HIV result.**

| | Total (n) | | HIV-negative | | HIV-positive | |
|---|---|---|---|---|---|---|
| | **n** | **%[1]** | **n** | **%[1]** | **n** | **%[1]** |
| **Total** | 10,076 | 100 | 7,815 | 77.4 | 2261 | 22.6 |
| **Age, years, mean (SD)** | 10,076 | 35.5 (7.8) | 7,815 | 34.6 (7.9) | 2261 | 38.6 (6.8) |
| **Age categories** | | | | | | |
| 14–24 years | 682 | 6.6 | 656 | 95.0 | 26 | 5.0 |
| 25–34 years | 4,108 | 40.9 | 3,514 | 85.7 | 594 | 14.3 |
| 35–44 years | 3,987 | 40.1 | 2,776 | 69.6 | 1,211 | 30.4 |
| ≥45 years | 1,299 | 12.5 | 869 | 65.8 | 430 | 34.2 |
| **Sex** | | | | | | |
| Male | 8,282 | 81.7 | 6,588 | 79.3 | 1,694 | 20.7 |
| female | 1,792 | 18.3 | 1,225 | 68.6 | 567 | 31.4 |
| **IDU duration in years, mean (SD)** | 10,050 | 15.4 (8.9) | 7,794 | 14.2 (8.8) | 2,256 | 19.4 (8.1) |
| **IDU duration categories, %:** | | | | | | |
| < 3 years | 764 | 7.2 | 716 | 93.3 | 48 | 6.7 |
| 3–5 years | 964 | 9.2 | 875 | 89.9 | 89 | 10.1 |
| 6–10 years | 1,678 | 16.3 | 1,464 | 87.4 | 214 | 12.6 |
| >11 years | 6,644 | 67.3 | 4,739 | 71.5 | 1,905 | 28.5 |
| **Education** | | | | | | |
| 9 or less school years | 1,714 | 17.0 | 1,264 | 73.5 | 450 | 26.5 |
| complete school or vocational school | 6,377 | 63.6 | 4,893 | 76.3 | 1,484 | 23.7 |
| technical school or bachelor | 1,130 | 10.9 | 950 | 85.6 | 180 | 14.4 |
| graduate (institute or university) | 849 | 8.5 | 703 | 82.9 | 146 | 17.1 |
| **Marital status** | | | | | | |
| single | 4,329 | 42.7 | 3,279 | 75.3 | 1,050 | 24.7 |
| married or have regular sexual partner | 5,745 | 57.3 | 4,534 | 78.9 | 1,211 | 21.1 |
| **Monthly income** | | | | | | |
| less or equal to minimum wage[2] | 4,499 | 43.0 | 3,271 | 72.5 | 1,228 | 27.5 |
| more than minimum wage[2] | 5,575 | 57.0 | 4,542 | 19.0 | 1,033 | 81.0 |
| **Lifetime homelessness history** | | | | | | |
| yes | 1,482 | 15.1 | 1,120 | 75.3 | 362 | 24.7 |
| no | 8,592 | 84.9 | 6,693 | 77.8 | 1,899 | 22.2 |
| **Lifetime incarceration history** | | | | | | |
| yes | 4,177 | 41.6 | 2,877 | 68.5 | 1,300 | 16.3 |
| no | 5,897 | 58.4 | 4,936 | 83.7 | 961 | 31.5 |
| **Recent incarceration history** | | | | | | |
| yes | 532 | 5.4 | 385 | 69.7 | 147 | 30.3 |
| no | 9,544 | 94.6 | 7,430 | 77.8 | 2,114 | 22.2 |
| **Risky injection behavior[2]** | | | | | | |
| yes | 5,387 | 56.0 | 4,059 | 75.8 | 1,328 | 24.2 |
| no | 4,689 | 44.0 | 3,756 | 79.5 | 933 | 20.5 |
| **Injection drug types[2]** | | | | | | |
| Opioids only | 6,393 | 63.3 | 4,788 | 74.6 | 1,605 | 25.4 |
| Stimulants only | 1,189 | 12.2 | 1,053 | 88.9 | 136 | 11.1 |
| Opioids and stimulants | 2,492 | 24.5 | 1,974 | 78.8 | 520 | 21.2 |
| **Injection frequency, days, median (IQR)[3]** | 10,056 | 25 (14–30) | 7,799 | 24.0 (12–30) | 2,257 | 30.0 (15–30) |
| **Injection frequency[2], categories, %:** | | | | | | |
| 1–3 days | 426 | 3.9 | 313 | 75.9 | 113 | 24.1 |
| 4–10 days | 1,665 | 16.5 | 1,372 | 81.9 | 293 | 18.1 |

(*Continued*)

**Table 1.** (Continued)

| | Total (n) | | HIV-negative | | HIV-positive | |
|---|---|---|---|---|---|---|
| | **n** | **%[1]** | **n** | **%[1]** | **n** | **%[1]** |
| 11–20 days | 2,459 | 24.7 | 2,011 | 82.1 | 448 | 17.9 |
| 21–30 days | 5,506 | 54.9 | 4,103 | 74.0 | 1,403 | 26.0 |
| **Risky sexual behavior[4]** | | | | | | |
| yes | 389 | 3.9 | 301 | 78.4 | 88 | 21.6 |
| no | 9,687 | 96.1 | 7,514 | 77.4 | 2,173 | 22.6 |
| **Lifetime syphilis history** | | | | | | |
| yes | 215 | 2.2 | 141 | 65.8 | 74 | 34.2 |
| no | 9,861 | 97.8 | 7,674 | 77.7 | 2,187 | 22.3 |
| **Received harm reduction services[5]** | | | | | | |
| yes | 4,912 | 48.0 | 3,583 | 72.8 | 1,329 | 18.4 |
| no | 5,164 | 52.0 | 4,232 | 81.6 | 932 | 27.2 |
| **Received medical (non-HIV specific) services[4]** | | | | | | |
| yes | 3,163 | 31.7 | 2,276 | 72.5 | 887 | 27.5 |
| no | 6,913 | 68.3 | 5,539 | 79.7 | 1,374 | 20.3 |
| **Received TB treatment[4]** | | | | | | |
| yes | 1,051 | 10.5 | 569 | 53.8 | 482 | 46.2 |
| no | 9,025 | 89.5 | 7,246 | 80.2 | 1,779 | 19.8 |
| **Lifetime OAT experience** | | | | | | |
| yes | 1,044 | 9.5 | 637 | 60.2 | 407 | 39.8 |
| no | 9,030 | 90.5 | 7,176 | 79.2 | 1,854 | 20.8 |
| **Current OAT** | | | | | | |
| yes | 467 | 4.3 | 260 | 55.7 | 207 | 44.3 |
| no | 9,609 | 95.7 | 7,555 | 78.4 | 2,054 | 21.6 |
| **Anti-HCV test result** | | | | | | |
| positive | 6,424 | 63.9 | 4,561 | 70.8 | 1,863 | 29.2 |
| negative | 3,652 | 36.1 | 3,254 | 89.1 | 398 | 10.9 |
| **ART duration[6,7]** | | | | | | |
| <6 months | | | | | 116 | 5.2 |
| 6 months and more | | | | | 701 | 29.9 |
| Not on ART | | | | | 1,444 | 65.0 |
| **Current ART non-adherence[6,7]** | | | | | | |
| Used ART during last 2 days | | | | | 748 | 32.3 |
| Used ART more than 2 days ago | | | | | 69 | 2.8 |
| Not on ART | | | | | 1,444 | 65.0 |

[1] Table presents population estimates that were adjusted for RDS study design

[2] Minimum wage in Ukraine as in December 2017 was 3,200 UAH (approximately $116)

[3] In the past month

[4] In the past three months

[5] In the past year

[6] ART duration and Current ART non-adherence was not calculated for HIV status awareness and self-reported ART uptake

[7] Column percentages

less likely to know about their HIV status, but the choice of drugs was not associated with any outcome. All health service uptake variables were significantly associated with HIV-status awareness and ART uptake, but not with viral suppression. Duration of ART and recent adherence were strongly associated with viral suppression.

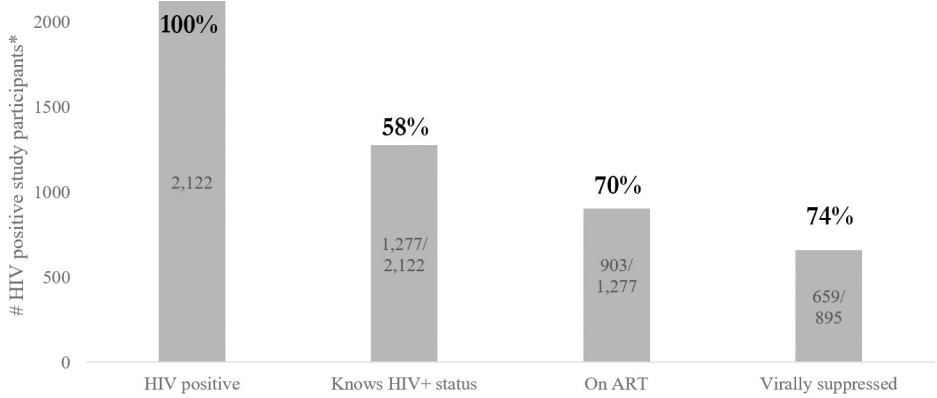

*The percentages are adjusted using RDS weights; numerators and denominators present unadjusted number of cases in the sample

**Fig 1. HIV treatment cascade among PWID in Ukraine based on 2017 IBBS data.** HIV-positive is defined as receiving HIV-positive result of two consecutive rapid-test. Awareness of HIV positive status is measured based on self-report information about HIV-positive previous test result. On ART is defined based on self-reported information about current ART uptake. Virally suppressed is defined as having viral load of <1000 HIV RNA copies per mL. based on lab test confirmation.

The results of the multivariable multilevel models of HIV-status awareness, ART uptake, and VL suppression are presented in Table 3. Graphs with fixed and random effects are presented in the (S1–S6 Figs).

Accessing harm reduction services in the last 12 months (AOR = 3.9, 95% CI: 3.1–4.9) and OAT lifetime participation (AOR = 3.0, 95% CI: 2.1–4.3) showed the strongest association with HIV status awareness. Experience of tuberculosis treatment in the last 12 months also had a significant association with HIV status awareness (AOR = 2.1, 95% CI: 1.6–2.9) and appeared as the strongest factor for ART uptake (AOR = 2.1, 95% CI: 1.5–2.9).

Several correlates remained significant for multiple outcomes in multivariable multilevel models. Lower monthly income and usage of non-HIV specific medical services remained significant for HIV status awareness and ART uptake.

Socio-demographic characteristics, such as female sex (AOR = 1.7, 95% CI: 1.3–2.2), longer IDU duration (AOR = 1.3, 95% CI: 1.1–1.4 for each additional 5 years) and seropositivity for HCV (AOR = 1.5, 95% CI: 1.2–2.1) remained significant only for HIV status awareness. Older age remained significant only for ART treatment uptake.

Self-reported ART treatment during the last 6 months or longer increased the odds of being virally suppressed (AOR = 1.7, 95% CI: 1.1–2.7). In contrast, poor adherence, indicated by not taking ART during the last two days, significantly decreased the odds of suppression (AOR = 0.4, 95% CI: 0.3–0.8).

In addition, each additional thousand HIV rapid tests distributed for the target population within each city during the survey year decreased the odds of HIV status awareness (AOR = 0.95, 95% CI: 0.89–0.99). PWID from the cities with a larger population size of people living with HIV were less likely to report ART uptake.

## Sensitivity analysis

We conducted a sensitivity analysis to assess the effect of exclusion from the analysis of participants who refused to report HIV status and were coded as missing values. HIV prevalence

**Table 2. Socio-demographic, behavioral, and service use characteristics of HIV-positive PWID in Ukraine, 2017.**

| | HIV status awareness | | | ART uptake | | | Viral suppression | | |
|---|---|---|---|---|---|---|---|---|---|
| | n | % (mean/SD)[1] | p-value | n | % (mean/SD)[1] | p-value | n | % (mean/SD) [1] | p-value |
| **Total** | 1,277/2,122 | 58.5 | | 903/1,277 | 69.8 | | 659/895 | 73.6 | |
| **Age, years, mean (SD)** | 2,122 | 39.1 (±6.4) | <0.001 | 1277 | 39.7 (±6.4) | <0.001 | 895 | 40.0 (±6.3) | <0.001 |
| **Age categories** | | | <0.001 | | | <0.001 | | | <0.01 |
| 14–24 years | 4/24 | 16.9 | | 1/4 | 28.6 | | 1/1 | 100 | |
| 25–34 years | 310/552 | 55.4 | | 187/310 | 61.8 | | 119/185 | 63.7 | |
| 35–44 years | 709/1,140 | 60.7 | | 513/709 | 70.6 | | 382/509 | 77.2 | |
| ≥45 years | 254/406 | 59.4 | | 202/254 | 78.4 | | 157/200 | 74.0 | |
| **Sex** | | | <0.001 | | | 0.891 | | | 0.933 |
| male | 921/1,588 | 56.6 | | 650/921 | 69.7 | | 474/643 | 74.0 | |
| female | 356/534 | 63.7 | | 253/356 | 69.9 | | 185/252 | 72.6 | |
| **IDU duration in years, mean (SD)** | 2122 | 20.4 (±7.4) | <0.001 | 1277 | 21.0 (±7.2) | <0.001 | 895 | 21.3 (±7.1) | <0.01 |
| **IDU duration categories, %:** | | | <0.001 | | | <0.001 | | | 0.544 |
| < 3 years | 13/46 | 26.9 | | 6/13 | 49.3 | | 3/6 | 54.5 | |
| 3–5 years | 26/82 | 33.0 | | 15/26 | 55.7 | | 10/15 | 60.2 | |
| 6–10 years | 92/201 | 46.0 | | 50/92 | 54.9 | | 36/49 | 75.1 | |
| >11 years | 1144/1788 | 61.9 | | 832/1,144 | 71.6 | | 610/825 | 73.9 | |
| **Education** | | | 0.191 | | | 0.766 | | | 0.331 |
| 9 or less school years | 239/426 | 54.6 | | 163/239 | 66.7 | | 113/159 | 71.4 | |
| complete school or vocational school | 847/1,395 | 58.7 | | 601/847 | 69.7 | | 439/597 | 73.5 | |
| technical school or bachelor | 103/164 | 62.3 | | 75/103 | 73.7 | | 54/75 | 69.5 | |
| graduate (institute or university) | 88/136 | 64.0 | | 64/88 | 74.1 | | 53/64 | 84.3 | |
| **Marital status** | | | 0.450 | | | 0.666 | | | 0.649 |
| single | 587/990 | 57.9 | | 419/587 | 70.9 | | 301/413 | 73.5 | |
| married or have regular sexual partner | 690/1,132 | 58.9 | | 484/690 | 68.8 | | 358/482 | 73.7 | |
| **Monthly income** | | | <0.001 | | | <0.001 | | | 0.428 |
| less or equal to minimum wage[2] | 777/1,161 | 65.3 | | 583/ 777 | 74.7 | | 431/578 | 74.1 | |
| more than minimum wage[2] | 500/961 | 50.8 | | 320/500 | 62.7 | | 228/317 | 72.7 | |
| **Lifetime homelessness history** | | | 0.503 | | | 0.243 | | | 0.463 |
| yes | 207/334 | 61.5 | | 139/207 | 67.3 | | 98/138 | 71.2 | |
| no | 1070/1,788 | 57.8 | | 764/1,070 | 70.3 | | 561/757 | 74.1 | |
| **Lifetime incarceration history** | | | 0.106 | | | 0.803 | | | 1.000 |
| yes | 758/1,229 | 59.4 | | 538/758 | 70.5 | | 141/533 | 73.8 | |
| no | 519/893 | 57.1 | | 365/519 | 68.7 | | 95/362 | 73.4 | |
| **Recent incarceration history** | | | 0.584 | | | 0.070 | | | 0.734 |
| yes | 77/133 | 58.5 | | 47/77 | 60.2 | | 33/46 | 75.5 | |
| no | 1200/1,989 | 57.5 | | 856/1,200 | 70.5 | | 626/849 | 73.5 | |
| **Risky injection behavior[2]** | | | 0.030 | | | 1.000 | | | 0.592 |
| yes | 737/1,265 | 55.5 | | 515/737 | 69.8 | | 380/511 | 74.0 | |
| no | 540/857 | 62.9 | | 388/540 | 69.8 | | 279/384 | 73.1 | |
| **Injection drug types[2]** | | | 0.085 | | | 0.054 | | | 0.811 |
| Opioids only | 927/1,507 | 59.3 | | 673/927 | 71.3 | | 487/673 | 72.8 | |
| Stimulants only | 65/123 | 55.0 | | 43/65 | 64.2 | | 33/43 | 79.1 | |
| Opioids and stimulants | 285/492 | 56.7 | | 187/285 | 66.2 | | 139/186 | 75.2 | |
| **Injection frequency, days, mean (SD)[3]** | 2122 | 22.2 (9.7) | <0.639 | 1277 | 22 (9.8) | 0.521 | 895 | 22 (10.0) | 0.487 |
| **Injection frequency[2], categories, %:** | | | 0.279 | | | 0.621 | | | 0.171 |
| 1–3 days | 69/107 | 68.9 | | 53/69 | 74.5 | | 43/53 | 83.9 | |

*(Continued)*

**Table 2.** (Continued)

| | HIV status awareness | | | ART uptake | | | Viral suppression | | |
|---|---|---|---|---|---|---|---|---|---|
| | n | % (mean/SD)[1] | p-value | n | % (mean/SD)[1] | p-value | n | % (mean/SD)[1] | p-value |
| 4–10 days | 177/275 | 64.5 | | 124/177 | 69.0 | | 89/124 | 74.0 | |
| 11–20 days | 241/416 | 58.0 | | 174/241 | 71.7 | | 137/174 | 75.8 | |
| 21–30 days | 788/1,320 | 56.6 | | 551/788 | 69.0 | | 390/543 | 72.1 | |
| **Risky sexual behavior[4]** | | | 0.818 | | | 0.257 | | | 0.709 |
| yes | 48/82 | 53.8 | | 38/48 | 80.9 | | 27/38 | 63.7 | |
| no | 1229/2,040 | 58.6 | | 865/1,229 | 69.4 | | 632/857 | 74.0 | |
| **Lifetime syphilis history** | | | 0.112 | | | 1.00 | | | 1.00 |
| yes | 50/72 | 66.3 | | 15/50 | 69.5 | | 9/35 | 72.0 | |
| no | 1227/2050 | 58.1 | | 359/1227 | 69.8 | | 227/860 | 73.7 | |
| **Received harm reduction services[5]** | | | <0.001 | | | <0.001 | | | 0.926 |
| yes | 970/1,260 | 75.6 | | 713/970 | 72.7 | | 519/706 | 73.8 | |
| no | 307/862 | 34.4 | | 190/307 | 60.7 | | 140/189 | 73.0 | |
| **Received medical (non-HIV specific) services[4]** | | | <0.001 | | | <0.001 | | | 0.289 |
| yes | 582/841 | 67.9 | | 446/582 | 75.6 | | 334/444 | 74.9 | |
| no | 695/1,281 | 52.4 | | 457/695 | 64.9 | | 325/451 | 72.4 | |
| **Received TB treatment[4]** | | | <0.001 | | | <0.001 | | | 0.935 |
| yes | 355/461 | 75.8 | | 292/355 | 81.0 | | 212/289 | 74.3 | |
| no | 922/1,661 | 53.6 | | 611/922 | 65.4 | | 447/606 | 73.3 | |
| **Lifetime OAT experience** | | | <0.001 | | | <0.001 | | | 0.071 |
| yes | 47/396 | 88.8 | | 282/349 | 81.4 | | 216/278 | 77.4 | |
| no | 798/1,726 | 52.1 | | 621/928 | 65.6 | | 443/617 | 72.0 | |
| **Current OAT** | | | <0.001 | | | <0.001 | | | 0.425 |
| yes | 190/201 | 94.6 | | 159/190 | 83.6 | | 120/157 | 77.2 | |
| no | 1087/1,921 | 55.0 | | 744/1,087 | 67.5 | | 539/738 | 72.9 | |
| **Anti-HCV test result** | | | <0.001 | | | 0.008 | | | 0.006 |
| positive | 1109/2,122 | 61.0 | | 310/1,109 | 71.0 | | 197/792 | 74.7 | |
| negative | 168/363 | 45.6 | | 64/168 | 61.3 | | 39/103 | 65.0 | |
| **ART duration[6]** | | | | | | | | | <0.001 |
| <6 months | | | | | | | 70/115 | 61.8 | |
| 6 months and more | | | | | | | 539/815 | 76.7 | |
| **Current ART non-adherence[6]** | | | | | | | | | <0.001 |
| Took ART during last 2 days | | | | | | | 571/746 | 76.3 | |
| Took ART more than 2 days ago | | | | | | | 38/69 | 54.1 | |

[1] Table presents population estimates that were adjusted for RDS study design

[2] Minimum wage in Ukraine as in December 2017 was 3,200 UAH (approximately $116)

[3] In the past month

[4] In the past three months

[5] In the past year

[6] ART duration and Current ART non-adherence was not calculated for HIV status awareness and self-reported ART uptake

among the participants who refused to report their status was 23.8%, compared to 24.0% in those who reported (p = 0.960). We have not found any evidence that HIV-positive participants who have reported about HIV status during the interview and these who decided not to report were different in their sex (p = 0.600) and age (p = 0.877). There was no statistical significant difference in the time period (HIV testing in the past 12 months) when the HIV tests

**Table 3. Final generalized linear mixed-effects models for HIV status awareness, ART uptake and viral suppression among PWID in Ukraine, 2017.**

| | HIV status awareness, n = 2,122 | | ART uptake n = 1,277 | | Viral suppression n = 895 | |
|---|---|---|---|---|---|---|
| | AOR (95% CI) | p-value | AOR (95% CI) | p-value | AOR (95% CI) | p-value |
| *FIXED EFFECTS Participant-level factors* | | | | | | |
| Aged 35–44 years (ref. <25 years)[1] | – | – | 1.7 (1.2–2.2) | 0.0013 | 1.5 (1.0–2.2) | 0.0606 |
| Aged 45 years and older (ref. <25 years)[1] | – | – | 2.3 (1.6–3.6) | <0.0001 | 1.7 (1.0–2.8) | 0.0425 |
| Female sex (ref.: male) | 1.7 (1.3–2.2) | 0.0002 | 1.1 (0.8–1.4) | 0.6270 | – | – |
| IDU duration, per 5 years | 1.2 (1.1–1.3) | <0.0001 | | | – | – |
| Monthly income: less than minimum wage (ref. more than minimum wage) | 1.4 (1.2–1.8) | 0.0034 | 1.5 (1.2–2.1) | 0.0017 | – | – |
| Received harm reduction services[2] (ref.: did not receive harm services) | 3.9 (3.0–5.2) | <0.0001 | – | – | – | – |
| Received medical (non-HIV specific) services[2] (ref.: did not receive services) | 1.4 (1.1–1.7) | 0.0266 | 1.6 (1.2–2.2) | 0.0005 | – | – |
| Lifetime OAT experience (ref.: no experience) | 3.1 (2.1–46) | <0.0001 | 2.0 (1.2–3.2) | 0.0018 | – | – |
| Received TB treatment[2] (ref.: did not receive treatment) | 2.2 (1.6–2.9) | <0.0001 | 2.1 (1.5–2.9) | <0.0001 | – | – |
| Positive anti-HCV test result (ref.: negative) | 1.5 (1.2–2.1) | 0.0027 | – | – | – | – |
| ART duration of 6 or more months (ref.: <6 months) | ‡ | ‡ | ‡ | ‡ | 1.7 (1.1–2.7) | 0.0033 |
| Current ART non-adherence | ‡ | ‡ | ‡ | ‡ | 0.4 (0.3–0.8) | 0.0193 |
| *City-level factors* | | | | | | |
| Number of PWID tested for HIV in the city / 1,000 | 0.95 (0.90–0.99) | 0.0430 | – | – | – | – |
| Estimated size of HIV-positive PWID population in the city /1000 | – | – | 0.97 (0.95–0.99) | 0.0036 | – | – |
| *RANDOM EFFECTS* | | | | | | |
| City (random intercept variance) | 0.661 | | 0.181 | | 0.083 | |
| Lifetime OAT experience (random slope variance) | 0.043 | | 0.419 | | ‡ | |
| Received medical (non-HIV specific) services[4] (random slope variance) | 0.090 | | 0.027 | | ‡ | |
| Received harm reduction services | 0.123 | | | | | |
| VPC in the null models | 0.217 | | 0.063 | | 0.031 | |

[1] Reference category for current ART uptake and VL suppression is <35 years old, because subsample of <25 years old was insufficient for this analysis.

[2] In the past year

"‡" variables that were not included into the model

"–"variables were excluded from the final model according to backward selection procedure

AOR–adjusted odds ratio

CI–Confidence Interval

p-value codes: '***', 0.001 '**' 0.01, '*' 0.05, '·' 0.1, ' ' 1.

were conducted for participants who tested HIV positive in the survey and hadn't reported their status in comparison to those who shared their HIV-status during the interview (37% vs 45%, p = 0.076). However, those participants who refused to report their HIV status were less likely to be virally suppressed (34% vs 49%, p<0.001), compared to those who shared their status during the interview.

## Discussion

### Cascade results

Overall, our findings confirm that significant gaps exist in the HIV treatment cascade among PWID in Ukraine. Only 58% of the estimated population of PWID living with HIV are aware of their status.

Compared to reports from other countries, the proportion of PWID aware of their HIV status was comparable to Mexico [48], but lower than in Central Asia or other European countries [7, 49].

We found a substantial 30% gap between HIV status awareness and ART initiation.

There is a paucity of published data on ART coverage among PWID globally, and the figures vary considerably across countries. According to the European Center for Disease Prevention and Control (ECDC) [49], UNAIDS [7] reports, and published reviews [8, 48], there is a stark difference between higher- and lower/middle-income countries (LMIC) in reaching 90-90-90 target among PWID. Compared to other LMIC globally and within the Eastern Europe and Central Asia (EECA) region, the 70% ART coverage among PWID who were aware of their HIV- positive status that we have found in this study is higher than the average. This also indicates that the gap between the number of diagnosed PWID and the number of treated patients is relatively smaller in Ukraine.

The final cascade indicator, viral load suppression (VLS), was achieved in approximately 74% of those PWID who reported being on ART. Additionally, 4.8% of HIV positive PWID with self-reported non-use of ART had been virally suppressed according to the lab results (4.5% among clients of harm reduction programs and 5.4% among non-clients). It might be attributed to the stage of the HIV-infection, interrupted ART uptake or misclassification of ART uptake due to the self-reported nature of data [50].

From the available data sources, we may conclude that the level of viral load suppression in Ukrainian PWID is modestly higher than among their peers in other LMIC, but lags far behind higher income countries, (88% among those treated in Western Europe) [7, 48].

## Implications for programming

This study confirms that HIV diagnosis is the biggest gap in the HIV care continuum among PWID in Ukraine. The national HIV program prioritized testing services in recent years [3, 51–53], but much remains to be done–an estimated 33,000 PWID remained unaware of their HIV-positive status. Our findings suggest that some subpopulations have restricted access to HIV testing services (HTS); these groups could benefit from focused testing strategies. These include older PWID, especially men, who are not actively using any other health or social services. Social network testing approaches, designed to deliver HTS to hard-to-reach PWID networks, have shown good results in Ukraine and could be scaled-up [54–56].

Compared to other countries with IDU-driven epidemics, the gap between HIV diagnosis and ART uptake in Ukraine is relatively small. This could at least partially be due to roll-out of OAT, covering 10,189 patients as of the end of 2017 [57] and case-management services focused on linkage of HIV positive PWID into care [40]. The role of OAT in facilitating access to ART is well established globally [58] and in Ukraine [59]. The case-management programmatic data from 2016 indicates that 75% of newly diagnosed PWID were enrolled into HIV care and 32% initiated ART [56]. Another more structured integrated intervention tested in Ukraine that was focused on linkage to HIV care and of OAT services maintenance was able to start and retain on ART 73% of participants in the experimental arm, compared to 36% in the standard of care arm at 6 months after enrollment [60]. Scale up of OAT and its close linkage with the HIV testing programs could be a key to closing this gap in the cascade.

The finding that TB treatment was an independent positive correlate of HIV status awareness and ART initiation confirms the importance of HIV/TB integrated services and suggests that this integration is having impact in Ukraine. Several studies support that integration of TB and HIV services can lead to substantial increase in HIV detection and ART initiation [61–63].

Our study also found disproportional anti-HCV prevalence for HIV-positive and HIV-negative PWID, which is in line with other studies from Ukraine (64, 65). It also found that PWID with anti-HCV positive results have increased odds of HIV status awareness. Thus, the finding

might result from a more intense history of testing and using other medical services among HIV-positive PWID. Therefore, further improvement in HCV detection and increasing HCV treatment coverage might have a positive effect not only for HCV but also for HIV diagnosis and treatment. With regard to viral load, the gap observed in our study is larger than would be expected among PLHIV given current ART efficacy [3]. A large body of evidence shows that with adequate support, PWID can achieve adherence and viral suppression levels comparable to other patients [64]. It is imperative to prioritize care and support services for PWID in the context of ongoing health care reform, to enable provision of these services through a wide range of community-level providers with governmental funding.

## Limitations

Our findings should be interpreted considering several important limitations.

First, calculation of cascade indicators excluded those participants who were unwilling to report their HIV status. These individuals were more likely to be HIV-positive, received HTS more frequently, and thus were more likely to know their status. This indicates an underestimation in our cascade estimates, but given the relatively small number in this group (n = 139, or 1.4%), the degree of underestimation is likely modest.

Two outcomes (status awareness and ART uptake) and all potential correlates in our analysis were collected by self-report, which is prone to report- and social-desirability biases. The IBBS study protocol did not include testing for metabolites of ARV, which could minimize this bias. A recent international study showed bi-directional bias in self-report of ARV use, and the Ukrainian subsample showed low discrepancy between the self-report and laboratory results [65]. This suggests that overall bias in ART uptake estimation should be minimal. ARV metabolite testing should be considered for the future studies to confirm this.

The sample recruited in IBBS may not be fully representative of the entire PWID population in Ukraine because the survey was conducted in larger cities where PWID populations are concentrated and because of potential selection biases. Besides the bias introduced by the RDS approach itself [66], its implementation in Ukraine could also vary by city and year, resulting in fluctuation of key indicators [25].

Lastly, the target study sample size for each RDS city was calculated to provide sufficient precision for measurement of HIV prevalence, but not ART uptake or viral suppression levels.

## Conclusions

The estimated HIV treatment cascade outcomes among PWID in Ukraine are comparable to other lower- and middle-income countries, and reveals substantial gaps in progress towards the 90-90-90 goals in this key population. Improving the care cascade among PWID must be a priority for the national and international stakeholders in HIV response in Ukraine, especially at the stage of HIV status awareness. Intensive HIV testing strategies targeting hard-to-reach PWID subgroups that are not currently accessing any other health services are needed to address this gap. Scale up of OAT and community-level linkage to care and ART adherence interventions are essential to improve ART coverage and viral suppression among PWID.

## Supporting information

**S1 Table. Characteristics of RDS recruitment in the study cities.**
(DOCX)

**S1 Fig. Fixed effects of multilevel multivariate models for "unaware about HIV positive status" outcome.**
(TIF)

**S2 Fig. Random effects of multilevel multivariate models for "unaware about HIV positive status" outcome.**
(TIF)

**S3 Fig. Fixed effects of multilevel multivariate models for "self-reported ART" outcome.**
(TIF)

**S4 Fig. Random effects of multilevel multivariate models for "self-reported ART" outcome.**
(TIF)

**S5 Fig. Fixed effects of multilevel multivariate models for "viral load suppression" outcome.**
(TIF)

**S6 Fig. Random effects of multilevel multivariate models for "viral load suppression" outcome.**
(TIF)

## Acknowledgments

We thank the Alliance Consultancy and IBBS data collection teams for their commitment and effort throughout the study.

We appreciate the support from the Public Health Center of the Ministry of Health Ukraine in planning and implementation of the study and chairing the National HIV Surveillance Group that had reviewed and approved the study protocol and instruments. Finally, we thank all study participants for taking part in the survey and willingness to share their experience.

## Author Contributions

**Conceptualization:** Yana Sazonova, Roksolana Kulchynska, Yuliia Sereda, Marianna Azarskova, Yulia Novak, Tetiana Saliuk, Marina Kornilova, Mariia Liulchuk, Charles Vitek, Kostyantyn Dumchev.

**Data curation:** Mariia Liulchuk.

**Formal analysis:** Yana Sazonova, Yuliia Sereda, Marianna Azarskova.

**Investigation:** Roksolana Kulchynska, Tetiana Saliuk, Charles Vitek.

**Methodology:** Yana Sazonova, Roksolana Kulchynska, Marianna Azarskova, Tetiana Saliuk, Marina Kornilova, Kostyantyn Dumchev.

**Project administration:** Yana Sazonova, Roksolana Kulchynska.

**Supervision:** Charles Vitek.

**Validation:** Roksolana Kulchynska, Mariia Liulchuk.

**Writing – original draft:** Yana Sazonova, Kostyantyn Dumchev.

**Writing – review & editing:** Roksolana Kulchynska, Yuliia Sereda, Marianna Azarskova, Yulia Novak, Tetiana Saliuk, Marina Kornilova, Mariia Liulchuk, Charles Vitek.

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
