## [Decision Letter · Decision Letter 0]

14 Dec 2020

HIV treatment cascade among People Who Inject Drugs in Ukraine

PONE-D-20-24973

Dear Dr. Sazonova,

We’re pleased to inform you that your manuscript has been judged scientifically suitable for publication and will be formally accepted for publication once it meets all outstanding technical requirements.

Kind regards,

Bharat S. Parekh, Ph.D.

Academic Editor

PLOS ONE

Additional Editor Comments (optional):

Reviewers' comments:

Reviewer's Responses to Questions

**Comments to the Author**

1. Is the manuscript technically sound, and do the data support the conclusions?

Reviewer #1: Yes

Reviewer #2: Yes

2. Has the statistical analysis been performed appropriately and rigorously? 

Reviewer #1: Yes

Reviewer #2: Yes

3. Have the authors made all data underlying the findings in their manuscript fully available?

Reviewer #1: No

Reviewer #2: Yes

4. Is the manuscript presented in an intelligible fashion and written in standard English?

Reviewer #1: Yes

Reviewer #2: Yes

5. Review Comments to the Author

Reviewer #1: I did not see that the raw data was made available. Overall, very well written manuscript and important addition to the literature on the progress on HIV treatment cascade to achieve UNAIDS 90-90-90 and potential service interventions to improve on gaps within the cascade

Reviewer #2: The publication “HIV treatment cascade among People Who Inject Drugs in Ukraine” demonstrate an important outcome of the 2017 Integrated Bio-Behavioral Surveillance (IBBS) study. IBBS is well established tool for public health strategy and actions planning and development for HIV infection control and prevention. Current study has been conducted in the Ukraine, country in many aspects representing so called WHO Euro region. Ukraine survey data on the HIV treatment cascade is a crucial tool to guide HIV prevention and treatment strategies in the region where scientifically based data are limited in several other countries such as Russian Federation, Turkmenistan and other.

There were estimated of more than 350,000 PWID in Ukraine. The HIV epidemic among PWID, one of the largest in Europe and remains concentrated in people who inject drugs. Several social and economic factors contributed to HIV outbreak magnitude including stigma, police violence and other factors associated with the different types of violence.

Presented study is incredibly important and relevant for the Ukraine and for the region in general.

Most significant finding “The biggest gap in the cascade was found in the first “90”, HIV status awareness: only 58% [95% CI: 56%-61%] of HIV-positive PWID reported being aware of their HIV-positive status. Almost 70% [67%-72%] of all HIV-infected PWID who were aware of their status

reported that they currently received antiretroviral therapy (ART)”. Reflect urgent necessity for HIV prevention and treatment improvement in Ukraine.

I do not have comments on methodology, data collection and analyses. That elements perform on international quality level and held understand for finding.

Limitation of the study well describe but does not constrained of the results.

Few comments on descriptive part I can suggest:

1. Was MoH or other government institutions involved into the survey or study was self-conducted by NGO and USA government agency only with the permission from MOH?

2. Because of results of previous IBBS rounds in Ukraine are available that may be important to discuss the changes in current study findings in comparison with previous results.

3. Because of study participants received the equivalent of 6 USD as compensation it is possible that most vulnerable part of PWID population were relatively more involved into the study. I would be good to mention.

6. PLOS authors have the option to publish the peer review history of their article (what does this mean?). If published, this will include your full peer review and any attached files.

Reviewer #1: No

Reviewer #2: **Yes: **Michael O. Favorov MD, PhD, DSc

---

## [Editor Report · Acceptance letter]

22 Dec 2020

PONE-D-20-24973 

HIV treatment cascade among People Who Inject Drugs in Ukraine 

Dear Dr. Sazonova:

I'm pleased to inform you that your manuscript has been deemed suitable for publication in PLOS ONE. Congratulations! Your manuscript is now with our production department. 

Kind regards, 

on behalf of

Dr. Bharat S. Parekh 

Academic Editor

PLOS ONE